# Recent Developments of ICG-Guided Sentinel Lymph Node Mapping in Oral Cancer

**DOI:** 10.3390/diagnostics11050891

**Published:** 2021-05-17

**Authors:** Ji-Hoon Kim, Minhee Ku, Jaemoon Yang, Hyung Kwon Byeon

**Affiliations:** 1Department of Otorhinolaryngology, National Health Insurance Service Ilsan Hospital, Goyang 10444, Korea; jihoonkim@nhimc.or.kr; 2Department of Radiology, Yonsei University College of Medicine, Seoul 03722, Korea; imechbio@yonsei.ac.kr; 3Department of Otorhinolaryngology-Head and Neck Surgery, Soonchunhyang University College of Medicine, Seoul 04401, Korea

**Keywords:** sentinel lymph node, near-infrared fluorescence imaging, oral cancer

## Abstract

Sentinel lymph node (SLN) biopsy has gained attention as a method of minimizing the extent of neck dissection with a similar survival rate as elective neck dissection in oral cancer. Indocyanine green (ICG) imaging is widely used in the field of surgical oncology. Real-time ICG-guided SLN imaging has been widely used in minimally invasive surgeries for various types of cancers. Here, we provide an overview of conventional SLN biopsy and ICG-guided SLN mapping techniques for oral cancer. Although ICG has many strengths, it still has limitations regarding its potential use as an ideal compound for SLN mapping. The development of novel fluorophores and imaging technology is needed for accurate identification of SLNs, which will allow precision surgery that would reduce morbidities and increase patient survival.

## 1. Introduction

Oral cancer is the sixth most common cancer, with over 0.6 million newly diagnosed patients every year worldwide [1]. Oral cancer has highly invasive features, and consequently, one-third of patients present with neck node metastasis at diagnosis, and the 5-year survival rate of these patients is only half that of those without metastasis [2]. Occult metastasis occurs in 30%–40% of clinically node-negative patients (cN0) who underwent elective neck dissection (END). In contrast, 60%–70% of patients with oral cancer undergo unnecessary neck dissection, which substantially increases the patient’s burden due to neck scarring, asymmetrical disfiguration, and functional deterioration [3].

Sentinel lymph node (SLN) biopsy has gained attention as a method of minimizing the extent of neck dissection with a similar level of survival rate as END. Research on SLN biopsy for oral cancers is lacking compared to that for other types of cancer. In a meta-analysis of the diagnostic evaluation of SLN biopsies performed on 3566 patients with early head and neck squamous cell carcinomas published in 2017, Liu et al. reported a clinically meaningful identification rate, sensitivity, and negative predictive value of 96.3%, 87.0%, and 94.0%, respectively [4]. However, SLN biopsies are not routinely used for treating head and neck cancers. The neck of patients still needs to be opened during the SLN biopsy procedure, resulting in a higher risk of injury to the marginal mandibular branch of the facial nerve than during the standard END procedure. In the case of oral cancers localized in the mouth floor, SLN biopsy with radiography has lower sensitivity due to the shine-through effect. Medical staff are exposed to radiation while using radioactive tracers, and the accuracy of intraoperative identification is frequently decreased due to the nature of the agent, resulting in poor spatio-temporal resolution [5,6]. Therefore, there is an unmet clinical need for simpler and safer methods of SLN biopsy, particularly when considering that some institutions may not have adequate radiation facilities, and medical providers are burdened by the idea of being exposed to radiation.

Indocyanine green (ICG) can be useful for identifying deep-lying lesions below the skin surface and can achieve high penetration during surgery [7]. ICG is the only intraoperative near-infrared (NIR) fluorophore approved by the Food and Drug Administration (FDA) [8]. ICG was initially used in liver function tests, as it is metabolized in the liver. It is now used as an off-label surgical tool during various oncologic surgeries to provide surgeons with additional real-time information [9,10]. It has been used to delineate surgical margins during primary tumor resection and to identify SLNs for detecting metastasis [11,12,13]. To overcome the drawbacks of the previous radiotracer-based SLN detection methods, the ICG-guided SLN mapping has been introduced. This technique is a real-time, high-resolution, non-ionizing, and inexpensive method that is easy to use and does not affect the surgical anatomy. This method also reduces the gap between the preoperative imaging and surgical operative findings [14,15,16,17,18]. In this study, we reviewed the existing literature on ICG-guided SLN mapping techniques in patients with oral cancer to examine the current techniques and their limitations and propose future directions for more accurate fluorescence image-guided surgical technology.

The search strategy of this literature review was as follows: PubMed and Google Scholar were used to search for references with the key terms “oral cancer”, “head and neck cancer”, “near-infrared fluorescence”, “sentinel lymph node”, “indocyanine green”, “image-guided surgery” and “molecular imaging”. Studies published in English until June 2020 were included. Clinical trials currently in progress on the ClinicalTrials.gov website were searched and organized using the key terms aforementioned.

## 2. Conventional Sentinel Lymph Node Biopsy 

Occult metastasis cannot be definitively diagnosed based on non-invasive imaging modalities, such as preoperative computed tomography (CT), magnetic resonance imaging, and positron emission tomography/CT. Therefore, END is performed for both staging and therapeutic purposes [19]. However, approximately 30% of patients who undergo END are later found to have pathologic lymph node metastases based on histologic results, indicating that 70% of these patients undergo unnecessary surgery [20,21]. Considering postoperative morbidity, they can be considered to have been over-treated. Proponents of the “wait-and-see strategy” for neck metastasis claim that initiating treatment once neck node metastasis is detected during follow-up of patients with cN0 oral cavity cancer does not significantly affect survival [22]. 

By definition, SLN refers to the first echelon lymph node where the tumor directly flows through the lymphatic duct. The concept of SLN has been proposed by Cabanas using lymphangiograms in patients with penile cancer, while intraoperative lymphatic mapping techniques have been introduced by Morton et al. in patients with melanoma [23,24]. Tumor metastasis via the lymphatic ducts is known to occur in serial patterns [25]. The likelihood of tumor metastasis can be predicted by assuming that lymph node metastasis first occurs in an SLN. Numerous studies have been conducted with the assumption that, in theory, if no cancer cells are detected in an SLN, then there is no metastasis in other higher-tier lymph nodes and lymph node dissection is unnecessary [26]. Therefore, intraoperative SLN biopsy has been accepted as an important technique for determining the extent of surgery in patients with various types of cancers [27].

## 3. Currently Used Fluorophore for Sentinel Lymph Node Biopsy

Existing methods of SLN biopsy use preoperative SLN identification using lymphoscintigraphy and intraoperative gamma probe detection with or without visible blue dye [28]. Isosulfan blue is a popular visible blue dye used in earlier SLN biopsy approaches. It has been found to selectively stain the lymphatic system in animal experiments and has been used in SLN biopsies for patients with breast cancer [29]. However, efforts to discover new dyes arose as 1%–3% of patients, in whom isosulfan blue was used, developed anaphylaxis, and there was also a shortage of dye [30]. Methylene blue shows a greenish-blue color in its crystal form and a blue color in its ionized form. It is rapidly excreted from the body as it is a low molecular weight compound with a peak excitation wavelength of 668 nm and a peak emission wavelength of 688 nm under NIR light [31]. Methylene blue is superior to isosulfan blue in terms of its supply, cost, and risk of side effects and has thus been widely used in clinical settings [32]. Studies have also shown that methylene blue is more effective than other dyes in detecting SLNs and that it effectively detects SLNs when used alone without any radiotracers [33,34]. Although it is easy to use, methylene blue may contaminate the operative field around the tumor. The risk of contamination is particularly high in oral cavity cancer as the primary tumor is located close to the lymph nodes in these regions.

## 4. Near-Infrared Fluorescence Imaging 

Visible light cannot penetrate deep into the body owing to the high rate of absorption by water and proteins within the body; therefore, it is challenging to detect it deep within biological tissues [35]. NIR light with a wavelength of 760–850 nm, where the rate of light absorption by water and blood is at its lowest, has the advantages of low absorption and reduced auto-fluorescence [36,37]. NIR has relatively high penetration depths for biological tissues (several mm) and has a high signal-to-background ratio, exhibiting a “white stars in a black sky” effect that facilitates target identification [38]. NIR fluorescence (NIRF) imaging is the most suitable method for image-guided surgery owing to its low risk of radiation exposure, light absorption in biological tissues, scattering, and auto-fluorescence [13]. 

ICG is the most commonly used NIR fluorophore which has a peak excitation length of 807 nm and a peak emission wavelength of 822 nm [39,40]. NIRF imaging using ICG achieves high skin penetration during surgery and is thus useful for identifying tumors or lymphatics located below the skin surface [41,42,43,44]. ICG is a water-soluble tricarbocyanine dye with a molecular weight of 774.96 Da and must be stored at 15–25 °C [45,46]. Once injected intravenously, it binds to plasma proteins and is selectively absorbed by the liver; it is not excreted by the kidneys but in bile [47,48]. As a relatively hydrophobic molecule, ICG rapidly binds to serum proteins and has a short half-life of less than 3 min [49]. It has negligible toxic effects on the body. Although allergic reactions to ICG, including urticaria and anaphylaxis, have been reported, ICG can still be safely used in clinical settings as the incidence of these reactions is quite low. The recommended dose of ICG for a liver function test is 0.5 mg/kg and using a dose higher than this is reported to increase the incidence of side effects [50]. ICG can be traced with the naked eye under visible light, but it cannot contaminate the operative field or interfere in the surgery because the strength of the fluorescence signal is not strong for the naked eye [51]. The retention and clearance rates of ICG in blood were measured to assess the liver and cardiac circulation. Since ICG allows for blood vessel imaging, it has also been used to acquire angiography for coronary or neurosurgeries [52].

The NIRF imaging system consists of a light source, a filter, and a detector, and is used to monitor fluorescent signals [53]. Since NIR is not visible to the human eye, a special imaging system that can excite NIRF signals in the operative field and collect the emitted photons, is necessary [54]. The NIRF imaging system receives a visible light image and a NIR image from each charged coupled device (CCD) camera simultaneously. It then outputs images that combine the signals from the two cameras in real-time. NIR fluorophores are injected into the body immediately before surgery, and the fluorescence emitted from the body when it is exposed to light from an external source is detected to produce and display an image on a monitor [55]. 

In clinical settings, Fluobeam (Fluoptics, France), Photodynamic Eye (PDE; Hamamatsu, Japan), SPY (Novadaq, Richmond, Canada), Mini-FLARE™ (Beth Israel Deaconess Medical Center, Boston, MA, USA), and Firefly technology (da Vinci Robotics System, Sunnyvale, CA, USA) HyperEye Medical System (HEMS; Mizuho Medical, Tokyo, Japan) are commonly used (Figure 1) [56]. Zhu et al. summarized the specifications of the existing imaging systems in detail [54]. Fluobeam, PDE, and HEMS are portable hand-held imaging systems, while SPY and Mini-FLARE™ are cart-based imaging devices. The PDE, which is most frequently used in clinics, uses a light-emitting diode (LED) with a fluence rate of 4.0 mW cm^−2^ as a light source and a CCD camera with a dynamic range of 8 bits; however, does not provide real-time integration of fluorescence images. Most fluorescence imaging systems use a laser diode with a fluence rate of 1–10 mW cm^−2^ or an LED and a CCD camera with a dynamic range of 8 or 12 bits. 

As the imaging systems vary in their fluorescence excitation wavelengths, the dynamic range of the camera, and the fields of view, it is difficult to quantitatively compare the performance of these systems. Furthermore, ICG is used off-label for purposes other than SLN identification, and standard measurement using ICG is not yet possible since the administration route and dosage of ICG are not yet standardized. Most systems that allow for monitoring of NIRF signals are expensive and require a certain amount of space owing to their large size. Additionally, NIRF signal monitoring must be performed after restricting the use of external light sources, such as operating lights, to prevent noise. Thus, the use of operating lights is limited during NIRF imaging. There has always been a clinical need for new imaging systems that can overcome these limitations. 

## 5. Recent Applications of ICG for Sentinel Lymph Node Biopsy in Oral Cancer 

### 5.1. ICG Concentration and Dose

ICG is peritumorally injected to identify lymphatic vessels and SLNs. High dose injection does not necessarily indicate a strong NIR signal. When the concentration of ICG is excessively high, the NIR signals may become weaker owing to concentration-dependent quenching [57]. An ICG concentration of 2.5–5 mg/mL is typically used in clinical settings. However, ICG at this concentration must still be diluted to a certain extent to generate a sufficient NIR signal [58]. When a high concentration of ICG is injected into the body, ICG may fail to produce NIRF signals as it does not become diluted, even if it reaches the SLN. Thus, it is important to determine the appropriate concentration of ICG in NIR imaging to effectively produce NIRF signals inside the body. Different concentrations of ICG are used depending on the cancer type and the purpose of using ICG. Although ICG produces sufficient NIR signals at a low concentration, recent studies have reported that ICG detects tumors more effectively at higher concentrations, suggesting the need to determine the optimal concentration and dose for different cancer types and purposes. Kong et al. reported that an ICG concentration of 0.1 mg/mL is appropriate for SLN detection in the stomach of a large animal model [59]. While ICG spread multi-directionally beyond SLNs when injected at a conventional concentration (5 mg/mL), it effectively visualized lymph nodes and lymphatic channels when injected at a diluted concentration (0.1 mg/mL). Mieog et al. attempted SLN mapping after injecting ICG mixed with human serum albumin (HSA) at varying concentrations in patients with breast cancer. They observed a drastic decrease in fluorescent signals due to quenching at concentrations above 400–800 mΜ [60]. Cho et al. reported excellent results following tumor resection after intravenous injection of a high concentration (5 mg/kg) in patients with brain tumor compared to the standard concentration [61]. Wang et al. conducted a study on 12 patients with oral cavity cancer to determine the optimal dose of ICG for detecting primary tumors [62]. They observed the highest contrast in fluorescent signals between tumor and normal tissues at 6 h after the injection of 0.75 mg/kg ICG. Since the aforementioned concentration was used for primary tumors, it cannot be used in SLN biopsy, requiring a peritumoral injection of ICG. The clinical studies of SLN biopsy in oral cavity cancer are summarized in Table 1 and Table 2. They used 0.5–2 mL of 2.5–10 mg/mL ICG during the procedures. No side effects were reported after injection of ICG at the concentrations and doses as illustrated in Table 1. Al-Dam et al. used an ICG dose of 0.5 mg/kg body weight in 2 mL, which is higher than the dose normally used in patients with oral cavity cancer and obtained reasonable SLN mapping results comparable to those of previous studies [63]. Further research on the optimization of the ICG dose for SLN detection is needed.

### 5.2. ICG Detection Timing 

It has been reported that SLNs can be detected approximately 10 min after a peritumoral injection for other cancer types [64,65]. In breast cancer, the time from incision to SLN identification is not long, as SLNs are observed immediately below the axillary skin. In the case of oral cavity cancer, however, SLNs are buried within the sternocleidomastoid muscle, making the transcutaneous visualization of SLNs difficult and time-consuming. Since NIR imaging systems are not yet commonly used in the field of oral cancer surgery, there is a somewhat steep learning curve for achieving an optimal balance when using an NIR camera. The incision made in which fluorescent signals are detected is inevitably large to expose the cervical area containing the fluorescent SLN. Researchers vary in their opinions regarding incision size and reaching a consensus will be a major challenge. However, in most cases, as shown in Table 1, the first SLN was detected within 10 min after injecting contrast agents when using an NIR imaging system. Van der Vorst et al. measured NIRF signals at 5, 10, 15, 20, 25, 30, 45, and 60 min after ICG-HSA injection during planned neck dissection in 10 patients with oral cancer [66]. They measured the NIRF signal draining into higher-tier nodes to determine the optimal detection time of the agent. They detected SLNs within 5 min of injection in seven patients and 10, 15, and 30 min in the remaining patients. The number of fluorescent SLNs significantly increased over time. Whereas the average number of fluorescent SLNs was 1.7 ± 0.8, 6.1 ± 3.5 SLNs were detected at 60 min after injection (*p* < 0.001). Although the ICG used in this study was not pure ICG, these results suggest that ICG spreads to the surrounding lymphatic system and emits fluorescence in higher-tier nodes over time. Kim et al. used Firefly^®^ techniques to identify SLNs during retroauricular robotic neck dissection in nine patients with oral cancer [67]. They injected ICG peritumorally 12 h before the surgery. Compared to previous studies, ICG was injected much earlier before surgery. They found an average of 3.4 fluorescent SLNs per patient. If the ICG had been sequentially drained out, it must have entered the systemic circulation, and there should have been no more SLNs that contained ICG. However, Kim et al. detected approximately the same number of SLNs, as in previous studies. Kong et al. hypothesized that ICG does not sequentially drain out of lymph nodes; however, consistently flows from the injection site to the first SLN; thus, fluorescent dyes can remain in the first SLN for a long period [58]. In cases where ICG is injected on the day of surgery, ICG is injected immediately before inducing general anesthesia or after skin flap elevation.

In these cases, SLNs are sometimes detected within 5 min of injection, and fluorescent SLNs can be observed up to 1 h after injection (Table 1). However, the time taken for ICG to reach the SLNs after peritumoral injection in the tongue is not exactly known. Although 10 min after ICG injection appears to be appropriate timing for the identification of fluorescent SLNs, further research on the biological properties of contrast agents is necessary for more effective SLN detection.

### 5.3. ICG Depth Penetration

ICG is known to have a skin penetration depth of approximately 1 cm. Several studies have attempted to transcutaneously detect fluorescent signals without making a cervical incision. Peng et al. attempted to transcutaneously visualize SLNs in the first five patients with oral cancer; however, was unsuccessful [68]. However, Nakamura et al. found that SLNs can be transcutaneously visualized by pressing on the skin below which SLNs are expected to be located using an HEMS device [69]. Christensen et al. also attempted transcutaneous visualization of SLNs; however, they only observed approximately 10% of all SLNs [70]. They found that a low BMI was significantly associated with the success of transcutaneous visualization (*p* = 0.05). When performing surgery on patients with a high BMI, surgeons must be aware that transcutaneous visualization may be difficult during SLN biopsy.

### 5.4. ICG Versus Radiotracer

Radiotracers have superior depth penetration relative to ICG. However, their major drawback is the shine-through effect. When a primary tumor is located on the floor of the mouth, radioactive signals spread near the injection site to generate strong background signals. The shine-through effect refers to when these background signals make it impossible to localize the SLNs around the level I area. In contrast, ICG effectively visualizes lymphatic flow without contaminating the operative field. NIRF imaging using ICG effectively detected SLNs near the primary tumor. Bredell examined in detail the pattern of lymphatic drainage near the submandibular glands in two patients using ICG alone [71]. Van den Berg et al. could not detect SLNs located near the injection site in four patients using a gamma probe; however, they found them using ICG [72]. However, van den Berg et al. could not excise the preoperatively identified SLNs located near the marginal branches of the facial nerves in two patients. Since the use of a gamma probe and NIR imaging was not able to distinguish between nerves and SLNs, they discontinued performing SLN biopsy and decided to perform a close follow-up of the patients. These results emphasize the need for contrast agents and high-resolution imaging systems that would allow for the precise identification of SLNs from normal anatomical structures. Borbón-Arce used hybrid tracers of ICG-^99m^Tc-nanocolloids to detect SLNs [74]. Two SLNs that were preoperatively detected in two patients with oral cancer were not detected during surgery owing to their proximity to the injection site. However, of the 22 additional SLNs detected during the surgery, 12 were detected near the injection site using NIRF imaging alone. Of the 25 patients, 6 had metastasis, and metastasis was present in the SLNs that were additionally detected during surgery in four of these patients. The additional lymph nodes were SLNs that were previously detected using a gamma probe and an NIR camera. These results demonstrate the importance of a multimodal approach in overcoming the limitations of the approaches using ICG or radiotracer alone. Nakamura et al. reported that using ICG alone or a combination of ICG and radiotracer reduces the SLN detection time more than using a radiotracer alone (19.8 ± 12.6 min vs. 30.6 ± 11.6 min) [69]. SLN metastasis was not observed in one patient in whom a radiotracer was used alone; however, lymph node metastasis was detected 1 year later (false negative issues). Christensen et al. detected 11 SLNs in only nine patients using NIRF imaging with a hybrid tracer during surgery [70]. Most of the SLNs were located close to the injection site. They were not detected on single photon emission CT (SPECT)/CT or using a gamma probe during surgery. Fluorescent SLNs can be easily differentiated from lymph node clusters. Micro-fluorescence scanning of tissue sections of fluorescent SLNs showed that SLNs emitted stronger fluorescence than non-SLNs. However, no metastasis was detected in the fluorescent SLNs. Based on these results, Christensen et al. suggested performing a biopsy of both fluorescent and radioactive SLNs. Honda et al. detected 29 SLNs during surgery, five of which were detected using intraoperative ICG alone [76]. Metastasis was detected in 5 of the 16 patients. All SLNs with metastasis were detected using preoperative CT lymphography and intraoperative ICG. Nodal recurrence occurred in 2 out of 11 patients who did not have metastasis during the follow-up period, with a mean length of 38 months (false negative). Locoregional recurrence did not occur in 5 patients who had metastasis. Honda proposed a combination of multimodal methods using preoperative CT lymphography and intraoperative ICG that does not use radiotracers. In a study comparing SLN mapping using radiotracer with ICG and that using ICG alone, several SLNs, particularly those near the injection site, was detected using ICG; however, not when using radiotracers. However, no biopsy results in which metastasis was detected in radiotracer-positive and ICG-negative SLNs have been reported. There is a possibility that a large amount of ICG flows into higher-tier nodes and emits stronger fluorescence in the adjacent lymph nodes than radiotracers. For the wider application of contrast agents, several issues, including false negatives, must be overcome. 

Currently, there is insufficient support for the use of ICG alone during SLN biopsy. There are few clinical trials of ICG-based tumors and SLN imaging for head and neck cancers (Table 3). There have been no comparative studies on ICG alone and existing radiotracers. Currently, two studies have compared images obtained using radiotracer and ICG, and those obtained using radiotracer only. Nakamura et al. published a retrospective study of NIR-guided SLN biopsy of 26 patients with head and neck skin cancer (melanoma, 19; squamous cell carcinoma, 10; mucoepidermoid carcinoma, 1) [78]. They compared the results between group A (*n* = 18) in which SLN biopsies were performed using radiotracers and isosulfan blue and group B in which radiotracers, isosulfan blue, and ICG were used (*n* = 12). In group A, 0.4–0.6 mL of ^99m^Tc-tin colloid was injected before the day of surgery, and SLNs were mapped using lymphoscintigraphy. On the day of the surgery, 0.6 mL of 2% isosulfan blue was peritumorally injected after anesthesia induction before the surgery. In group B, 0.6 mL of 0.5% ICG was additionally injected at the same site following the injection of ^99m^Tc-tin colloid and isosulfan blue. An SLN biopsy was then performed using a handheld gamma probe and a customized NIR camera. A higher detection rate was achieved in group B than in group A (95% vs. 83%). Additionally, while no recurrence occurred in group B during the follow-up period (18–84 months), nodal recurrence occurred in patients with negative SLNs in group A (false negative issues). Stoffels et al. conducted a prospective randomized clinical trial of 40 patients [79]. They compared a group in which a traditional radiotracer, ^99m^Tc-nanocolloid, was used (group B, *n* = 20) and a group in which a hybrid tracer, ICG-^99m^Tc, was used (group A, *n* = 20). The contrast agents were injected 21–23 h before surgery, and SLNs were preoperatively identified using lymphoscintigraphy and SPECT/CT. SLN biopsy was performed using a gamma probe and a Fluobeam system during surgery. No significant differences in the intraoperative SLN detection rate, SLN detection time, or the number of metastatic SLNs were found between groups A and B. A total of 36 SLNs were identified in group A, 20 of which were preoperatively detected, 30 of which were intraoperatively detected using a gamma probe, and 36 of these were detected using ICG. Most recently, Yokohama et al. reported the results of a prospective, multicenter, phase II clinical trial in 18 oral cancer patients. Before surgery, lymphoscintigraphy was performed, and ICG was injected during the surgery. SLN mapping was attempted using a PDE device while neck compression was performed using a plastic cone. ICG guided SLN mapping using this simple compression method confirmed high concordance with the radiotracer guided SLN mapping [77]. 

There are two major disadvantages of ICG-guided SLN mapping in oral cancer. ICG has a low skin depth penetration rate within 10 mm, and is rapidly migrated through the lymph nodes, hence fluorescent SLN is more likely to be found than when using a radiotracer. Due to the anatomical characteristics of the cervical region, there is limitation of the transcutaneous visualization of lymph nodes using ICG fluorescence imaging alone [77]. Furthermore, false-negative nodes may occur due to changes in the lymphatic system following surgical intervention, such as skin elevation and muscle retraction, and rarely due to skipping metastasis or reduced sensitivity of the imaging system. Due to these drawbacks, it is recommended to use ICG in conjunction with radiotracers in an SLN biopsy for patients with oral cancer; the use of ICG alone is not recommended [80]. A comparative study of ICG and radiotracers is needed to establish a basis for the use of ICG alone.

## 6. Advanced Imaging Contrast Agents

Although ICG has many strengths, it still has limitations regarding its potential use as an ideal compound for SLN mapping. Since ICG is amphipathic, it easily aggregates and lacks stability in an aqueous solution and undergoes a drastic reduction in the quantum yield [81]. ICG easily flows through SLNs due to its small diameter (≤5 nm). This results in reduced SLN mapping efficiency as fluorescence is not maintained for a long time and is detected in several higher-tier lymph nodes [82,83]. Furthermore, ICG does not target cancer specifically [84]. ICG exhibits passive extravasation through loose blood vessels around cancer cells and accumulates within cancer cells. This phenomenon is referred to as the enhanced permeability and retention effect [85,86]. Since ICG does not actively and specifically target specific types of cancer cells, there is always the possibility of false negatives. Therefore, research is actively being conducted to increase SLN accumulation and retention rates of ICG. Without any modifications to its chemical structure, ICG does not easily bind to other substances [39]. Although various chemical bonds can be used in pre-clinical trials, those aside from a simple mixing method cannot be used for ICG in clinical settings. In the case of ICG-^99m^Tc-nanocolloid, a hybrid tracer, the two compounds do not always stay bound together, indicating that free ICG may enter higher-tier nodes. Khullar et al. developed a hybrid compound using HSA as a method to retain the in vivo stability and fluorescence of ICG and used it in SLN mapping [87]. They managed to prevent the in vivo aggregation of ICG and achieve a three-fold increase in fluorescence yield in lymph nodes owing to the formation of nanoparticles with a diameter of 7.3 nm, thereby increasing the efficiency of SLN mapping. There have been attempts to overcome the limitations of ICG, such as self-aggregation, short half-life, and non-specific targeting, by binding ICG to a nanoparticle consisting of a polymer and an inorganic matrix. Tsuchimochi et al. successfully identified SLNs in the neck of a rat using a tracer created by loading ^99m^Tc and ICG, which were normally mixed, into polyamidoamine-coated silica nanoparticles [88]. Mok et al. designed ICG nanoparticles surrounded by hyaluronic acid (HA) [89]. This contrast agent does not emit fluorescence due to quenching under normal conditions. Overexpression of HAdases due to cancer or lymph node metastasis leads to the degradation of HA by HAdases, which allows ICG to emit fluorescence. Mok et al. verified the effectiveness of their contrast agent in a nude mouse model of breast cancer and a lymph node model. Polymer-, lipid-, or silica-based and magnetic ICG nanoparticles have also been developed [90]. Although research on these contrast agents is still at the proof-of-concept stage, they offer various platforms that can overcome the limitations of ICG (Figure 2). Further research is needed until these contrast agents can be used in clinical practice.

## 7. Remarks

According to the SLN biopsy guidelines in patients with oral cancer, ICG is recommended as an adjunct to the radioactive tracer, particularly for the cancer of the floor of the oral cavity [80]. Radiotracer-guided SLN biopsy remains the standard for SLN localization. However, surgical consensus guidelines agree on the need for new technological developments, and this challenge is expected to bring about surgical innovations for SLN biopsy. ICG has been widely used and provides surgeons with real-time visual information; however, there is the main drawback of low penetration rates. To overcome this problem, a fluorescent and radioactive hybrid tracer has been used as a new alternative. A hybrid of ICG and ⁹⁹mTc-nanocolloid can identify deep-lying lesions, demonstrating a superior SLN mapping rate of 95% or more than the existing blue dye in various cancers. However, more accurate tumor targeting, and targeted fluorophores are needed to avoid missing on occult metastatic lesions and for precise tumor margin assessment. Various molecular imaging agents have been developed to target specific biomarkers. The effects of these agents vary significantly from cancer to cancer and can express different types within the same tumor. In particular, the head and neck cancers have high intra-tumoral heterogeneity; thus, even if effective molecular imaging agents have been developed in animal models, there are limitations in applying them directly to the human body. The contrast agent developed in preclinical studies still requires many procedures, such as obtaining an appropriate dose for actual clinical translation, identifying bio-distribution patterns in the body, and evaluating toxicity in humans. Additionally, a high-resolution imaging system that can properly implement this process visually is also essential.

Van Dam et al. implemented first-in-human intraoperative real-time tumor imaging using a fluorescent agent targeting folate receptor-α overexpressed in ovarian cancer [92]. This technology selectively targets tumors, leading to innovative developments in surgical oncologic imaging. In general, images taken before surgery may differ from reality at the time of surgery. By visualizing the tumor in real-time, the extent of the surgery can be determined and adjusted according to the patient’s current situation by identifying the pattern of the tumor spread and whether there is an invasion of the surrounding normal tissues. Rosenthal et al. are the leading research group conducting clinical trials of targeted fluorescent agents in the head and neck area. Phase I dose-escalation studies were conducted by conjugating cetuximab with the fluorescent dye IRDye800CW [93,94,95,96]. A high tumor-to-background ratio and increased sensitivity (92.7%) were confirmed by lymph node imaging, confirming that it provides surgical guidance by accurately allowing visualization of primary tumors and metastatic lymph nodes. Additionally, by developing a fluorescent tumor mapping method through a Phase I study of IRDye800CW-labelled panitumumab, it was confirmed that important information on tumor margin assessment can be provided to surgeons during surgery [96,97,98,99]. Since fluorescence guide surgery is not yet a standardized surgical technique, discussion of standard procedures through large-scale multicenter studies is needed. Long-term follow-up studies through Phase II–III studies are required.

Given the anatomical location, it is less difficult to access the oral cavity than other parts of the body. Additionally, the potential metastatic lymph node distribution pattern of oral cancer is somewhat predictable in comparison to different types of cancers, and, relatively, the lymph nodes are located at a layer not very deep in the skin. It is more advantageous to standardize safety and efficacy evaluation if the contrast agent is administered intravenously. However, in the case of oral cancer, studies regarding topical administration techniques that can reduce systemic toxicity are necessary. Therefore, it is plausible to perform fluorescence-guided surgery if a well-established fluorescence imaging-based SLN mapping technique is adequately utilized. However, translational research for molecular imaging technology using contrast agents is required in the clinical phase. In particular, the development of novel fluorophores and imaging technology will allow for minimally invasive procedures and precision surgery that would reduce morbidities and increase patient survival.

## Figures and Tables

**Figure 1 diagnostics-11-00891-f001:**
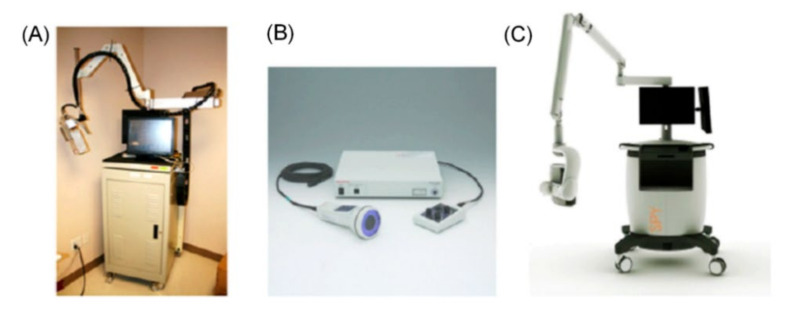
Commonly used near-infrared fluorescence imaging devices. (**A**) Frequency-domain photon migration (FDPM) imager, (**B**) Photodynamic eye, (**C**) SPY device. Adapted from Zhu B et al. Br. J. Radiol. 2015 Jan;88(1045):20140547 [54].

**Figure 2 diagnostics-11-00891-f002:**
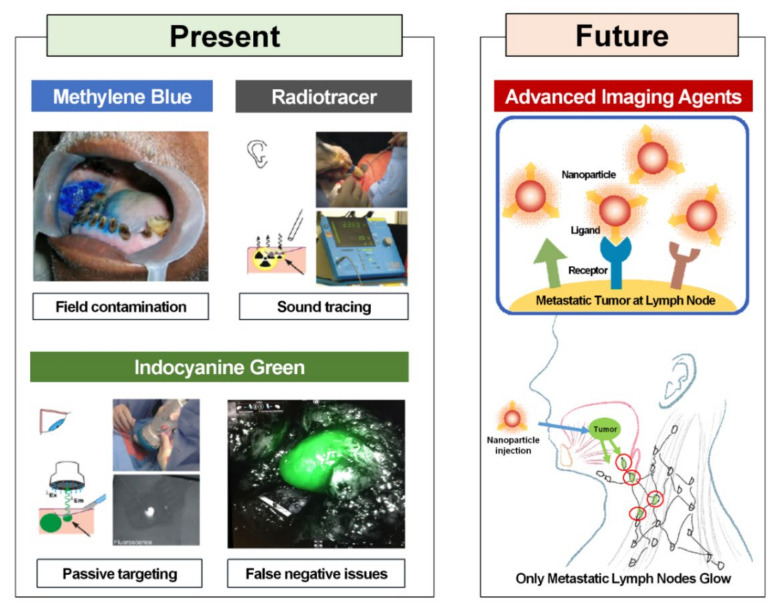
Present and future of sentinel lymph node biopsy surgical technics in oral cavity cancer. Reproduced from [72,91] with permission of Springer Nature.

**Table 1 diagnostics-11-00891-t001:** Clinical studies of ICG guided sentinel lymph node biopsy in oral cancer.

Author	Stage	Population	NIR Imaging Device	Tracer	ICG Dilution Solvent	ICG Concen-Tration (mg/mL)	ICG Dose (ml)	Injection Route	ICG Injection Time	Time to SLN Identification after Injection
Bredell (2010) [71]	TxN0	8 (5: OSCC, 3: Maxilla)	PDE	ICG alone	sterile water	10	ICG: 1	Peritumoral at least 5P	After induction of anesthesia	30 min initially, down to 5 min or less in the latter cases
van den Berg (2012) [72]	T1-2N0	14	HEMS	ICG-^99m^ Tc nanocolloid	sterile water	5	total 0.4 mL of median of 77 (range 67–94) MBq hybrid tracer	Peritumoral 3-4P	3–19 h before surgery	NA
Iwai (2012) [73]	TxN0	1	HEMS	ICG alone	NA	5	ICG: 0.5–1	Peritumoral 4P	After induction of anesthesia	Within several minutes
van der Vorst (2013) [66]	T1-2N0	10 (8: OSCC, 2: OPC)	Mini-FLARE	ICG:HSA	sterile water	2.5	1.6-mL of 500 μM ICG:HSA	Peritumoral 4P	After flap elevation	5, 10, 15, 20, 25, 30, 45 and 60 min
Borbón-Arce (2014) [74]	T1-2N0	25 (9: OSCC, 16: Melanoma)	PDE	ICG-^99m^ Tc nanocolloid	sterile water	5	total 0.4 mL median of 85 MBq (range 66–158 MBq) hybrid tracer	Peritumoral 3-4P	3–24 h before surgery	NA
Murase (2015) [75]	T1-2N0	16	PDE	ICG+ ^99m^ Tc-tincolloid	sterile water	5	ICG: 0.4 (0.4ml of 74MBq ^99m^Tc–tin colloid)	Peritumoral	After induction of anesthesia	NA
Peng (2015) [68]	T1-2N0	26 (19: OSCC, 7: OPC)	OMIONS	ICG + MB	NA	5 (MB: 10)	ICG: 1 (MB: 1.5)	Peritumoral 4P	Before skin incision	NA
Nakamura (2015) [69]	T1-2N0	19 (15: OSCC, 2: OPC, 2:HPC)	HEMS	^99m^Tc-tin colloid (*n* = 13), ICG + ^99m^Tc-tin colloid (*n* = 4), ICG (*n* = 2)	sterile water	2.5	ICG: 0.5 (1.0 mL of ^99m^ Tc-tin colloid)	Peritumoral 4P	SLN detection at 15 min after ICG injection	ICG or ICG + RI: 19.8 ± 12.6 min RI alone: 30.6 ± 11.6 minutes
Christensen (2016) [70]	T1-2N0	30	Fluobeam 800	ICG-^99m^ Tc nanocoll	sterile water	5	total 0.2 mL of hybrid tracer (55 MBq at same day, 110 MBq at day before surgery)	Peritumoral 4P	NA	from skin incision to skin closure: average 39 min
Al-Dam (2018) [63]	T1-2N0	20	PDE	ICG	sterile water	higher	0.5 mg/kg in 2 mL	Peritumoralat least 5P	After flap elevation	8.1 min (range 1–22)
Honda (2019) [76]	T1-2N0	18	HEMS/ PDE	ICG	sterile water	5	ICG: 2mL	Peritumoral	After flap elevation	1 or 2 min after injection
Kim (2020) [67]	T1-2N0	9	Da Vinci Robotic system Firefly	ICG	sterile water	2.5	ICG: 2 mL	Peritumoral 4P	12 h before surgery	NA
Yokohama (2020) [77]	T2-3N0	18	PDE	ICG	NA	2.5	NA	Peritumoral 4P	During surgery	10 min after injection, transcutaneous SLN detection

NIR, near-infrared; SLN, sentinel lymph node; Ref., references; OSCC, oral squamous cell carcinoma; OPC, oropharyngeal cancer; HPC, hypopharyngeal cancer; ICG, Indocyanine green; P, point; MB, methylene blue; NA, Not Applicable.

**Table 2 diagnostics-11-00891-t002:** Efficacy of previous clinical studies.

Author	Preoperative Imaging Modality	Number of Preoperative Localized SLNs	Number of Intraoperative Radioactive SLNs	Number of Intraoperative Fluorescent SLNs	Number of Patient with Detected SLNS	Number of Patient with Metastatic SLNs	Recurrence (Number of Patients)	Type of Surgical Procedure	Number of Patient with False Negative
Bredell (2010) [71]	NA	NA	NA	1–5 per patient (average 3)	8/8 (100%)	1 (12.5%)	NA	Biopsy	NA
van den Berg (2012) [72]	LSG followed by SPECT/CT	41	43	47	14/14 (100%)	1 (7.1%)	NA	Biopsy	NA
Iwai (2012) [73]	CT lymphography	NA	NA	NA	NA	NA	NA	Biopsy	NA
van der Vorst (2013) [66]	NA	NA	NA	17 (average 1.7 ± 0.8 per patient)	10/10 (100%)	3 (30%)	NA	Planned neck dissection	1
Borbón-Arce (2014) [74]	LSG followed by SPECT/CT	67	87	86	25/25 (100%)	6 (24%)	NA	Biopsy	0
Murase (2015) [75]	LSG followed by SPECT/CT	25	28	35	16/16 (100%)	2 (12.5%)	1 (in positive SLN): DOD-11 months, 1 (in negative SLN): DOC-24 months (3 years follow up)	Biopsy	0
Peng (2015) [68]	NA	NA	NA	88 (average 3.4 per patient)	26/26 (100%)	4 (15.4%)	NA	Planned neck dissection	0
Nakamura (2015) [69]	LSG	31	31	ICG alone: total 3 LNs, ICG + RI: average 3 LNs, RI alone: average 2 LNs	19/19 (100%)	2 (10.5%) ICG alone: none ICG + RI: 1 RI alone: 1	1 (RI-alone): nodal recurrence 1 year later	Biopsy(+)-> neck dissection	1
Christensen (2016) [70]	LSG followed by SPECT/CT	68	83	94	30/30 (100%)	6 (20%)	NA	Biopsy	0
Al-Dam (2018) [63]	NA	NA	NA	39 (average 1.95 per patient)	20/20 (100%)	8 (40%)	4: regional relapse (2.4 years follow up)	Planned neck dissection	4
Honda (2019) [76]	CT lymphography	25 (16/18 pts)	NA	29	16/16 (100%)	5 (31.3%)	2	T1-2: Biopsy(+)-> neck dissection, Advanced T2: Planned neck dissection	2
Kim (2020) [67]	NA	NA	NA	31	9/9 (100%)	2 (22.2%)	None (4 years follow up)	Planned neck dissection	0
Yokohama (2020) [77]	LSG with or without SPECT/CT	NA	63	67	18/18(100%)	5/18 (27.7%)	5	Biopsy(+)-> neck dissection	0

LSG, lymphoscintigraphy; RT-PCR, reverse-transcriptase polymerase chain reaction; SCCA, squamous cell carcinoma antigen; NED, no evidence of disease; DOC, died of other cause; DOD, died of disease; NA, Not Applicable.

**Table 3 diagnostics-11-00891-t003:** Clinical trials of ICG guided tumor and sentinel lymph node imaging in head and neck cancer.

ClinicalTrials.Gov Identifier	Start	No. of Patients	Target	Timing	Dose	Primary and Secondary Outcome	Country
NCT02027831	2013	10	All patients requiring neck dissection with or without resection of the primary head and neck cancer	Intravenous injection before the surgery	0.25 mg/kg	distribution of ICG in the normal and pathological lymph nodes	Belgium
NCT02640170	2015	500	Resectable solid tumors (lung, breast, kidney, parathyroid, prostate, stomach, head and neck etc.)	NA	NA	monitor the rate of recurrence in patients who undergo cancer surgery. (prospective design)	USA
NCT02920216	2016	10	Salvage surgery for recurrence of head and neck cancer in irradiated area	Intravenous injection before the surgery	0.25 mg/kg	Sensitivity of ICG in irradiated area and surgical margins	France
NCT02997553	2017	744	ICG guided SLN biopsy compared with the ^99m^Tc guided SLN biopsy in patients with cancers and subjected to surgery. (breast, head and neck, melanoma, cervix, rectum etc.)	Intravenous injection	2.5 mg/mL	Non-inferiority of ICG guided SLN biopsy	France
NCT03745690	2018	20	Head and neck cancer	Intravenous injection the day before surgery	NA	safety profile of high-dose ICG, the efficacy of high-dose ICG to identify cancer compared to surrounding normal tissue	USA

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
