# Peer review of "Recent Developments of ICG-Guided Sentinel Lymph Node Mapping in Oral Cancer"

_diagnostics, 2021, doi:10.3390/diagnostics11050891_

Round 1
Reviewer 1 Report
Manuscript title “Recent Developments of ICG-guided Sentinel Lymph Node Mapping in Oral Cancer”
Comments:
The manuscript is describing the Recent Developments of ICG-guided Sentinel Lymph Node Mapping in Oral Cancer which is useful for many surgeons who treat Oral cancer.
I believe that this manuscript is worthy of publication in this journal. However, it is necessary for the following points to be addressed prior to publication.
- Please add the disadvantage of ICG-guided Sentinel Lymph Node Mapping in Oral Cancer in the following.
- ICG Depth Penetration
- Number of SLN detected by ICG procedure more than Radiotracer
3.Due to the anatomical characteristics of the main SLN covered by the sternocleidomastoid muscle, there are many limitations of the transcutaneous visualization of SLN using ICG fluorescence imaging alone.
Has any relevant published work not been cited?
Authors should cite and discuss the paper: J Yokoyama, Y Hasegawa, M Sugasawa, A Shiotani, Y Murakami, S Ohba, and N Kohno. Long term‑follow‑up multicenter feasibility study of ICG fluorescence‑navigated sentinel node biopsy in oral cancer. Mol Clin Oncol. 2020 Oct;13(4):41.
This paper analyzes the sensibility and specificity of indocyanine green, compared with Radiotracer. This paper can indicate to overcome these problems of ICG fluorescence procedure. It should be cited and discussed in detail.
Reviewer 2 Report
Although this review paper is interesting, there are some points that should be corrected.
This review paper describes the current situation about sentinel lymph node mapping using indocyanine green in oral cancer.
The contents of this review article are interesting.
However, there are some points should be revised.
- In the line of 74th, “positron emission tomography-CT” should be “positron emission tomography/CT”.
- In the lines of 83rd and 84th, the contents of the following sentence is not correct: “It was first discovered by Morton et al., who found that the lymphatics surrounding a tumor moved to the SLN in patients with melanoma [23].”
The concept of sentinel lymph node had been proposed by Dr. Cabanas. He studied penile cancer and published this idea in the following paper. Cabanas RM: An approach for the treatment of penile carcinoma. Cancer 39 (2): 456-466, 1977. Dr. Morton would be the first man that clearly established the concept of sentinel node.
- In the line of 117th, “NIRF” should be spelled out.
- In the lines of 132nd and 133rd, the contents of the following sentence is not correct: “Since ICG cannot be seen with the naked eye under visible light, it does not contaminate the operative field and does not interfere with the surgery [51].”
ICG can be traced with the naked eye under visible light. It looks green although the strength of the signal is not strong. This sentence should be modified.
- In the line of 151st, “Israel Beth Deaconess Medical Center, USA” should be “Beth Israel Deaconess Medical Center, Boston, MA, USA”.
- In the line of 177th, the hyphen in this line is colored red. This should be colored black.
- In the lines of 363rd and 364th, the idea of “enhanced permeability and retention effect” was first appeared in the following paper: “Matsumura Y, Maeda H: A new concept for macromolecular therapeutics in cancer chemotherapy: mechanism of tumoritropic accumulation of proteins and the antitumor agent smancs. Cancer Res 46(12 Pt 1): 6387-6392, 1986. This landmark study should also be cited.
- In the line of 380th, “99m” of “99mTc” should be described by the superscript form.
- In the line of 396th, the sources of the guidelines should be cited.
- The references 35 and 39 are the same paper. This duplication should be corrected.
- The references after 62nd are described by an incorrect style. The journal names are wrongly described.
